# Histone Variant H2A.J Is Enriched in Luminal Epithelial Gland Cells

**DOI:** 10.3390/genes12111665

**Published:** 2021-10-22

**Authors:** Christophe E. Redon, Zoe Schmal, Gargi Tewary, Adèle Mangelinck, Régis Courbeyrette, Jean-Yves Thuret, Mirit I. Aladjem, William M. Bonner, Claudia E. Rübe, Carl Mann

**Affiliations:** 1Developmental Therapeutics Branch, Center for Cancer Research, National Cancer Institute, Bethesda, MD 20892, USA; redonc@mail.nih.gov (C.E.R.); aladjemm@mail.nih.gov (M.I.A.); bill3801@gmail.com (W.M.B.); 2Department of Radiation Oncology, Saarland University Hospital, 66421 Homburg/Saar, Germany; zoe.schmal@uks.eu (Z.S.); gargi.tewary@uks.eu (G.T.); 3Institute for Integrative Biology of the Cell (I2BC), CEA, CNRS, Université Paris-Saclay, 91190 Gif-sur-Yvette, France; adele.mangelinck@gmail.com (A.M.); regis.courbeyrette@cea.fr (R.C.); jean-yves.thuret@cea.fr (J.-Y.T.)

**Keywords:** histone variant H2A.J, luminal breast cancer, luminal prostate cancer, cancer biomarker, *H2AFJ*, H2AJ, estrogen-responsive genes

## Abstract

H2A.J is a poorly studied mammalian-specific variant of histone H2A. We used immunohistochemistry to study its localization in various human and mouse tissues. H2A.J showed cell-type specific expression with a striking enrichment in luminal epithelial cells of multiple glands including those of breast, prostate, pancreas, thyroid, stomach, and salivary glands. H2A.J was also highly expressed in many carcinoma cell lines and in particular, those derived from luminal breast and prostate cancer. H2A.J thus appears to be a novel marker for luminal epithelial cancers. Knocking-out the *H2AFJ* gene in T47D luminal breast cancer cells reduced the expression of several estrogen-responsive genes which may explain its putative tumorigenic role in luminal-B breast cancer.

## 1. Introduction

The diploid human genome of 6 billion base pairs of DNA is packaged into cell nuclei 10 µm in diameter by histone binding and condensation into nucleosomes. Nucleosomes comprise 145–147 bp of DNA wrapped around octamers containing two of each of the four core histone families—H2A, H2B, H3 and H4. A minimum of 20 bp of DNA bound to the linker histone H1 extends between nucleosomes. In addition to their roles in compacting DNA and organizing chromatin structure, histones are regulators of gene expression, DNA replication and DNA repair amongst others. Histones are defined as either canonical or variant based on their expression, their functional roles and genomic assignment (single gene versus several gene copies in clusters). Canonical histones are highly expressed in S phase to allow repackaging of the genome coupled to DNA replication. Transcriptional activation of the canonical histone genes and selective stabilization of their mRNAs account for this S-phase enrichment [1]. Canonical histone mRNAs typically terminate with a stem-loop sequence that is recognized by the SLBP (stem-loop binding protein) instead of a poly A sequence [2]. Binding of the stem-loop sequence by SLBP is required for maturation, stabilization, and translation of the canonical histone mRNAs. SLBP is degraded outside of S-phase and this in turn destabilizes the histone stem-loop RNAs. In contrast to the canonical histones, genes coding for histone variants are expressed independently of DNA replication and their mRNAs terminate in polyadenylation sequences. Histone variant expression is thus particularly important in differentiated post-mitotic cells. However, there have been few systematic studies of tissue-specific expression of histone variants. Histone variants often have specialized functions [3]. Mammals have eleven H1 variants, nine H2A variants, five H3 variants, at least two testis-specific H2B variants, and one H4 variant [4].

The core histone H2A has the largest number of replication-independent variants. The H2A.X, H2A.Z, and macro-H2A variants have been extensively studied in recent years, but H2A.J has received less attention and its precise roles are poorly understood. H2A.J seems to be specific to mammalian cells and differs from canonical H2A only by a valine in place of alanine at position 10 of the mature protein species, and the last seven amino acids that contain a potentially phosphorylatable SQ motif [5]. H2A.J has been implicated in the expression of inflammatory genes in senescent fibroblasts [5,6]. H2A.J over-expression has also been implicated in luminal-B breast cancer tumorigenesis [7,8] and in the chemotherapeutic resistance of colorectal [9], hepatic [10], and glioblastoma cancers [11]. We performed a global survey of H2A.J expression in human and mouse tissues to obtain initial indications regarding potential tissue-specific functions.

## 2. Materials and Methods

### 2.1. Cell Culture and Reagents

T47D cells were obtained from the NCI-60 cancer cell line repository. Cells were cultured in RPMI-1640 medium supplemented with GlutaMAX (Thermo-Fisher Scientific 61870036), 10% fetal bovine serum, and penicillin (50 U/mL) + streptomycin (0.2 mg/mL) (Sigma P0781, Saint-Quentin-Fallavier, France) in a 5% CO_2_ incubator. For experiments involving estrogen deprivation and feedback, T47D cells were first cultivated for three days in phenol red-free RPMI-1640 medium (cat# 11835030, Life Technologies; Grand Island, NY, USA) containing 5% charcoal-dextran-treated FBS (cat# A33821-01, Life Technologies; Grand Island, NY, USA) (CDFBS). Cells were harvested after two days of treatment with 10 nM estradiol and washed twice with PBS prior to whole-cell protein extraction. β-estradiol was purchased from Sigma Aldrich (cat# E8875-250MG, Saint Louis, MO, USA) and prepared as a 20 mM stock solution in DMSO. Prior to use, β-estradiol was diluted to 10 µM in ethanol (×1000 solution).

### 2.2. Crispr-Cas9 Knock-Out of the H2AFJ Gene

The H2AFJ gene deletion was performed using the CRISPR/Cas9 system as previously described [12]. Briefly, the H2AFJ-specific guide RNA sequence (GAGGTGATCATGTCCGGTCG) was chosen from http://crispr.mit.edu (Synthego tool, last accessed on 18 October 2021), and inserted into the pX330 vector (Addgene, Cambridge, MA, USA). For the donor plasmid, the left and the right homology arms were PCR-amplified from human primary fibroblast genomic DNA. The two sets of primers targeting human H2AFJ are listed as follows: for the upper region, H2AFJ_trgt_FW1 (gcgttcaacattacatcacg) and H2AFJ_trgt_RV2 (gtaaggtactccaacaccgc); and for the bottom region, H2AFJ_trgt_SI_FW2 (gtctggGTCGACgcgcgtgacaacaagaagac) and H2AFJ_trgt_ApI_RV1 (gtaataGGGCCCacgactaaaggcgctgc), respectively. Both arms along with antibiotic resistance ORF (hygromycin) were inserted between EcoRI and ApaI sites of the PCR2.1 vector (Life Technologies; Grand Island, NY, USA). Selection of H2AFJ-KO clones was performed by growth in the presence of 400 µg/mL hygromycin. Expression of a cDNA encoding Flag-HA-H2A.J from the tet-ON promoter was achieved by infecting H2A.J-KO cell lines with the previously described pTRIPz lentiviral-based construct [5].

### 2.3. Immunohistochemical Detection of H2A.J in Mouse and Human Tissue Sections

Characterization of the H2A.J antibody and its use in immunohistochemical staining was previously described [5,13]. Similar results were obtained with an anti-H2A.J antibody commercialized by Active Motif (catalog # 61793) that was also prepared to the unique C-terminus of H2A.J.

Normal (catalog # BN242 and BN243) and breast cancer (catalog# BR963a) human tissue microarrays were obtained from US Biomax (Rockville, MD, USA). Quantification of the breast cancer microarray was done by classifying the staining on a scale from 0 (no staining) to 4 (highest staining) and then calculating the mean for ER-positive and ER-negative biopsies. Mouse tissue sections were prepared as previously described [5].

### 2.4. Immunofluorescent Detection of H2A.J in Cultured Cells

Cancer cells in culture were fixed with 4% paraformaldehyde and prepared for immunofluorescence with 1/500 dilution of H2A.J antibody as previously described [5,14].

### 2.5. Western Blots

For H2A.J Western blots, whole cell extracts were prepared by resuspending trypsinized cells in SDS-PAGE sample buffer and heating at 95 °C for 15 min. Viscosity of genomic DNA was reduced by sonication with a probe sonicator. Extracts were electrophoresed in SDS-12% polyacrylamide gels and proteins were transferred to nitrocellulose membranes for detection of proteins with the Li-Cor Odyssey imager. The rabbit H2A.J antibody was used at a dilution of 1/500 in Odyssey blocking buffer. For TFF1 Western blots, cells were washed quickly twice with PBS, directly solubilized in denaturing sample buffer supplemented with 5% β-mercaptoethanol (200 µL/1 million cells), boiled for 5 min and then subjected to SDS polyacrylamide gel electrophoresis. Proteins were electrotransferred to 0.45 μm PVDF sheets (cat# IPVH304F0, Millipore, MA, USA) for immunodetection with the following primary antibodies: TFF1, Cat# 15571S, Cell Signaling technology (Danvers, MA, USA), and β-actin (cat# A2228-200UL, Sigma-Aldrich, Saint Louis, MO, USA).

### 2.6. Isolation of H2A.J-KO Mice

Cyagen injected fertilized eggs of C57BL/6-N mice with a TALEN nuclease targeting the 5′end of the *H2afj* gene, and used these eggs to impregnate surrogate mothers. Sequencing of tail DNA from the resulting pups identified a male mouse with a heterozygous 7-nucleotide deletion at the beginning of the *H2afj* gene (∆12–18 starting at 1 from the ATG) that would code for a frameshifted protein with a premature stop codon. The hetererozygous mouse was mated with a B6-N female that yielded male and female mutant offspring. Mating of male and female heterozygotes led to the isolation of male and female homozygous H2afj-∆7 mutant mice. We were unable to detect H2A.J protein by Western blots, immunofluorescence, and mass spectrometry analyses of organs and MEFs derived from these mutant mice [15]. Tissue sections from these H2A.J-KO mice were used to show the specificity of our IHC detection of H2A.J in C57BL/6-N WT mice. A full phenotypic characterization of the H2A.J-KO mice will be published at a later date.

### 2.7. RNA-seq of T47D and T47D-H2A.J-KO Cell Lines

RNA was isolated from 3 replicates each of T47D cells and 2 different Crispr-Cas9 H2A.J-KO clones (KO-1 and KO-18) under conditions of proliferation and after treatment with 5 µM tamoxifen for 7 days. RNA was purified with a Machery-Nagel Nucleospin RNA Plus kit. Poly A+ RNAs were purified and converted into cDNA for sequencing with the Illumina TruSeq stranded mRNA kit. The library preparation and 43 nucleotide paired-end DNA sequencing were performed by the I2BC high-throughput DNA sequencing platform.

### 2.8. Bioinformatic Analyses of T47D RNA-seq Data

The fastq sequences were trimmed with Cutadapt, quality controlled with FastQC, and mapped to gencode.v32.transcripts.fa.gz (GRCh38 transcriptome) using salmon1.1 with decoy sequences [16]. The resulting quant files were then used for exploratory data analysis and to identify differentially-expressed transcripts at the gene level using tximeta [17], DESeq2 [18], csaw [19], edgeR [20], limma [21], and voom [22]. The salmon quant files and R scripts are available in Zenodo (DOI: 10.5281/zenodo.5519652). The fastq files have been deposited at the EBI Array Express Database under accession number E-MTAB-10861.

### 2.9. GTEx, TCGA and CCLE Transcriptome Analyses

GTEx v. 8 data were downloaded and analyzed with R code deposited in Zenodo (DOI: 10.5281/zenodo.5414141). H2AFJ expression data in human cancer tissues (Cancer Genome Atlas, TCGA) and human cancer cell lines (the Broad Cancer Cell Line Encyclopedia, CCLE) were downloaded from the CBioportal (https://www.cbioportal.org/, last accessed on 18 October 2021) and the Cell minerCDB (https://discover.nci.nih.gov/rsconnect/cellminercdb/, accessed on 18 October 2021) portals, respectively. Graphs were generated using GraphPad Prism.

### 2.10. Single-Cell RNA-seq Analyses

Previously published human and mouse mammary gland single-cell RNA-seq data [23,24] were analyzed using a modified version of bioinformatic scripts made available by Saeki et al. (DOI: 10.5281/zenodo.4674274). The modified code is available in Zenodo (DOI: 10.5281/zenodo.5517186). Single-cell RNA-seq of normal human prostate cells [25] was analyzed with the Web site provided by the Strand Lab (https://strandlab.net/sc.data/hu.pd.deep/, accessed on 19 July 2021). Single-cell RNA-seq data of normal mouse prostate cells [26] was interrogated through the Broad Institute Single Cell Portal (https://singlecell.broadinstitute.org/single_cell/study/SCP859/, accessed on 20 July 2021).

### 2.11. Cancer Survival Plots

The breast cancer survival plots were generated by the Breast Cancer Gene-Expression Miner [27] v4.7 Web site (http://bcgenex.ico.unicancer.fr/BC-GEM/GEM-Accueil.php?js=1, accessed on 20 July 2021).). The R script analysis of prostate cancer survival as a function of H2AFJ expression was deposited in Zenodo (DOI: 10.5281/zenodo.5517186). The remaining cancer survival data plots were generated by OncoLnc [28].

## 3. Results

### 3.1. Immunolocalization Shows Enrichment of H2A.J in Luminal Epithelial Cells of Multiple Glands in Mice and Humans

We previously noted some organ-specific differences in H2A.J protein levels in the mouse by mass spectrometry [5]. Here, we used immunohistochemistry (IHC) with H2A.J-specific antibodies to visualize H2A.J expression within different cell types in adult mouse and human tissue sections (Figure 1). We previously demonstrated the specificity of these antibodies [5,13], and we provide further confirmation by comparing tissue sections from WT and H2A.J-KO mice and showing an example of competition of the H2A.J signal by an excess of the C-terminal H2A.J-specific peptide that was used as the immunogen to prepare the antibody (Appendix A). We observed striking cell-type specific variation in H2A.J expression (Figure 1 and Appendix A). In particular, we saw increased levels of H2A.J in luminal epithelial cells of multiple glands, including mammary, prostate, thyroid, pancreas, stomach, bladder, and salivary glands. Similar staining was observed for the corresponding human and mouse tissues.

We next mined transcriptome databases to see whether differences in H2A.J protein levels by immunohistochemistry were correlated with differences in *H2AFJ* RNA levels in different tissues. First, poly A+ RNA-seq data in transcripts per million (TPM) were analyzed for 27 different tissues from the GTEx (Genotype-Tissue Expression) Portal (Figure 2A) [29]. This subset contained TPM data for 9 to 2642 recently deceased generally non-diseased human donors depending on tissue type (Appendix A). These data were not normalized to an external RNA spike-in, and so can only be used as a qualitative indication of relative RNA expression between tissues.

The testis showed the highest *H2AFJ* expression followed by prostate, adipose tissue, and breast. The relatively high *H2AFJ* RNA levels in the prostate and breast are consistent with the strong luminal epithelial staining in the prostate and breast tissue (Figure 1A–D). Adipocyte nuclear staining was also seen in the mouse mammary tissue (Appendix A). IHC of the mouse testis showed strong staining mainly of the spermatogonia (Appendix A). Other tissues showing strong luminal epithelial IHC staining such as the pancreas, salivary glands, bladder, and stomach, had somewhat lower levels of *H2AFJ* RNA levels. Mass spectrometry indicated that the mouse liver and kidney contained significant amounts of H2A.J [5], despite containing relatively modest levels of *H2AFJ* RNA according to the human GTEx data (Figure 2A). IHC showed that H2A.J was expressed in hepatocyte nuclei and enriched in nuclei of the kidney cortex (Appendix A). Finally, tissues such as the heart and brain showed low levels of *H2AFJ* RNA, and IHC indicated that there were low levels of H2A.J in cardiomyocytes and the cerebrum (Appendix A). Overall, these data suggest that there is tissue-specific variation in *H2AFJ* RNA expression that contributes to differences in H2A.J protein expression, but some type of post-transcriptional regulation may be required to explain why tissues such as the liver and kidney appear to contain significant levels of H2A.J protein [5] despite relatively low *H2AFJ* RNA levels.

For these same 27 tissues, we also compared GTEx data for H2AFJ relative to the 5 other somatic cell H2A histone variant genes (*H2AFX*, *H2AFV*, *H2AFZ*, *H2AFY*, and *H2AFY2*, encoding H2A.X, H2A.Z1, H2A.Z2, macroH2A.1 and macroH2A.2 respectively) (Appendix A). The mean and standard deviation of the median TPM values for these RNAs in all tissues show that the *H2AFZ* gene is expressed at the highest overall levels, followed by *H2AFV*, *H2AFJ*, and *H2AFX*, and then the two macroH2A genes (Figure 2B). H2AFZ and H2AFJ also showed the greatest tissue-specific variation in RNA levels (Figure 2B and Appendix A).

Whole tissue RNA and protein determinations do not account for cell-type specific expression within tissues. The IHC images show cell-type specific variation within various tissues, such as the strong labeling of luminal epithelial cells. IHC of brain slices showed that whereas most cell types including neurons expressed low levels of H2A.J, a minor cell population expressed H2A.J at much higher levels (unpublished data). We examined single-cell RNA databases for human and mouse tissues to check for cell-type specific variation in *H2AFJ* expression.

### 3.2. Single-Cell RNA-seq Data Show High H2AFJ RNA Expression in Mammary and Prostate Luminal Epithelial Cells

We used previously published single-cell RNA-seq data to examine *H2AFJ* expression in mammary epithelial cells in female humans and mice [23,24]. 24,280 mammary epithelial cells from 4 healthy women aged 17–36 years old were grouped into 3 clusters by UMAP (uniform manifold approximation and projection): a basal-myoepithelial cluster (labeled B) and 2 luminal cell clusters (L1 and L2) (Figure 3A,B) [24]. The L1 cluster was characterized as secretory type luminal cells and the L2 cluster as hormone-responsive luminal cells [24]. Comparing *H2AFJ* expression with that of marker genes for basal (*KRT14*), L1 (*SLPI*), L2 (*ANKRD30A*), and both L1 and L2 (*KRT18*), revealed that *H2AFJ* is preferentially expressed in both L1 and L2 clusters compared to basal/myo-epithelial cells (Figure 3A). This result is evident as well from a heat map of gene expression comparing *H2AFJ* expression with that of the top 10 marker genes for each of these clusters (Figure 3B). The preferential expression of *H2afj* in luminal mammary cells is conserved in the mouse as seen with an integrated dataset of 50,407 epithelial cells from 5 independent studies leading to the definition of 6 cell clusters: a mammary stem/progenitor cell cluster (MaSc-pro), a basal cell cluster (Basal), a luminal hormone-sensitive cell cluster (L-Hor), a progenitor cluster for the L-Hor cells (LH-pro), a luminal alveolar cell cluster (L-Alv), and a progenitor cluster for the L-Alv cells (LA-pro) [23]. *H2afj* expression was higher in all the luminal clusters compared to the MaSc-pro and Basal clusters, with the highest expression in the hormone-sensitive luminal clusters (Figure 3C). These data suggest that the enhanced H2A.J protein levels in luminal epithelial mammary cells (Figure 1A,B) is due at least in part to increased levels of H2afj RNA expression in these cells.

We further examined *H2AFJ* expression in 28,606 single normal prostate cells from 3 men aged 18–31 years old [25]. As for the mammary glands, *H2AFJ* RNA expression was highest in luminal epithelial cells and low in basal epithelial cells of the prostate (Appendix A). Luminal expression of *H2AFJ* was similar to the *KLK3* gene encoding the prostate cancer antigen that is a marker of prostate luminal epithelial cells. However, whereas *KLK3* is highly restricted to luminal epithelial cells (Appendix A), there is significant expression of *H2AFJ* in other prostate cell clusters with epithelial club cells that have an immune-secretory transcriptome [25] showing the next highest level of expression (Appendix A). Single-cell RNA-seq of the mouse prostate gland [26] also showed highest expression of H2afj in luminal epithelial cells (Appendix A). Thus, differential expression of H2A.J in luminal mammary and prostate cells is correlated with higher levels of *H2AFJ* RNA in these cells.

### 3.3. H2A.J Is a Marker of Luminal Breast and Prostate Cancer Cells

*H2AFJ* was reported to be over-expressed in some cancers [7,8,9,11], and interrogation of the Cancer Genome Atlas (TCGA) database showed that prostate and breast cancer expressed high levels of *H2AFJ* mRNA (Figure 4, left panel). *H2AFJ* RNA was also highly expressed in many cancer cell lines derived from glandular tissues such as the breast, pancreas, lung, colorectum, thyroid, esophagus, stomach, and prostate according to the Broad Cancer Cell Line Encyclopedia (CCLE) (Figure 4, right panel).

Interestingly, immunofluorescence experiments showed that H2A.J was expressed at much higher levels in breast and prostate cancer cell lines of luminal cell origin [30,31] compared to basal or non-luminal cell origin [30,31,32] (Figure 5A–C), and Western blotting confirmed this result for the breast cancer lines (Figure 5D). Since we observed high expression of H2A.J in normal breast and prostate luminal epithelial cells compared to basal cells (Figure 1 and Figure 2), this observation suggests that luminal cancer cells retain the high H2AFJ expression of their cell of origin and that H2A.J expression might represent an additional marker of luminal epithelial cancer cells.

Consistent with previous reports [7], breast cancer transcriptome databases [33] show that *H2AFJ* RNA expression was highest in luminal-B breast cancer cells, followed by luminal-A, HER2-E, Normal breast-like, and Basal-like (Figure 6).

We performed IHC staining of a tissue microarray composed of 76 malignant core tissue samples (58 ER-negative breast cancer samples and 18 ER-positive samples) (Figure 7 and Appendix A). Luminal breast cancers are typically ER-positive whereas basal breast cancers are typically ER-negative [30]. Quantification of the H2A.J signal showed significantly higher levels of H2A.J staining for the ER-positive breast cancers that is consistent with the higher *H2AFJ* RNA levels for luminal versus basal breast cancers (Figure 6). This result suggests that H2A.J represents a novel IHC marker for luminal ER-positive breast cancers.

### 3.4. Single-Cell RNA Analaysis of H2AFJ Expression in Prostate Cancer Cells

A recent single-cell RNA-seq study of 36,424 cells from 12 patients with prostate cancer identified 16 epithelial cancer cell clusters [34]. Most prostate cancers show a luminal-type gene expression profile [34]. We found that *H2AFJ* expression in prostate cancer epithelial cells was enriched in luminal-type epithelial cells with a profile that was very similar to the *KLK3* gene encoding the prostate specific antigen (PSA) that is used as a marker of luminal prostate cancer (Appendix A). In particular, violin plots indicated that both *H2AFJ* and *KLK3* had the highest expression levels in prostate cancer epithelial clusters 3, 4, and 9, and showed the lowest expression in the basal cell epithelial cluster (Appendix A). Plotting the mean values of *H2AFJ* versus *KLK3* RNA levels for the prostate cancer clusters showed a high correlation coefficient of 0.88 (Appendix A). Thus, increased *H2AFJ* RNA expression is characteristic of both breast and prostate luminal-type cancers.

### 3.5. H2AFJ Expression in Relation to Breast Cancer Survival

Breast cancers with a high expression of *H2AFJ* had a slightly higher survival rate compared to breast cancers that expressed low levels of *H2AFJ* (Figure 8). Since luminal breast cancer cells express *H2AFJ* at higher levels than basal breast cancer cells, this survival difference may reflect the more aggressive nature of basal versus luminal breast cancer [35]. Separating the breast cancer survival data according to PAM50 intrinsic molecular subtypes, high *H2AFJ* expression was correlated with a slightly elevated survival for all subtypes except luminal A breast cancer for which it showed significant negative correlation with survival (Appendix A).

### 3.6. H2AFJ Expression in Relation to Survival for Other TCGA Cancer Types

Prostate cancer patients with higher than median *H2AFJ* expression tended to show slightly better survival compared to patients with lower than median *H2AFJ*, but this difference did not reach statistical significance because there were only 10 death events in this TCGA cohort of 498 primary prostate cancer patients (Appendix A). Interrogation of all TCGA cancers in the OncoLnc survival database [28] indicated that *H2AFJ* significantly affected the survival of 4/21 cancer types: high *H2AFJ* expression was associated with poorer survival for brain lower-grade glioma (LGG), kidney renal-cell carcinoma (KIRC), and skin cutaneous melanoma (SKCM), but higher survival for bladder urothelial carcinoma (BLCA) (Appendix A). Thus, the effect of H2AFJ expression on cancer survival depends on the cancer type.

### 3.7. Transcriptomic Analysis of T47D H2A.J-KO Cells

To search for a possible function for H2A.J in luminal epithelial cells, we chose to inactivate the *H2AFJ* gene in T47D luminal breast cancer cells using the CRISPR/Cas9 technology. Two H2A.J-KO clones (KO1 and KO18) lacking detectable expression of H2A.J by Western blotting and immunofluorescence were characterized (Appendix A). We sequenced polyA+ RNAs from 3 replicates of the 2 independent H2A.J-KO cell lines and the parental T47D cells under conditions of proliferation and after treatment with 5 µM tamoxifen for 7 days. Tamoxifen is a selective estrogen-receptor modulator that is used as a first-line treatment to inhibit the growth of breast cancers expressing the estrogen receptor [35]. Treatment of the parental T47D cells and the 2 H2A.J-KO cell lines induced a similar inhibition of cell growth (Appendix A). Euclidean distance heat maps of the RNA-seq data indicated that samples were first separated by the conditions of tamoxifen-treated versus untreated proliferating cells and next by genotype (T47D versus H2A.J-KO) (Figure 9).

We first performed differential gene expression analysis for T47D cells treated with tamoxifen versus proliferating cells. 2462 genes showed differential expression (1279 up-regulated and 1183 down-regulated genes with FDR < 0.01) for tamoxifen-treated versus proliferating T47D cells (Figure 10A and Appendix A). Tamoxifen is thought to induce the proliferative arrest of mammary epithelial cells by modifying the expression of genes regulated by the estrogen receptor transcription factor ESR1 [36]. We performed gene set enrichment analyses (GSEA) using the c2 gene signature and Hallmark collections from the Molecular Signatures Database (MSigDB) [37] to identify gene sets whose expression is affected by tamoxifen treatment. The gene sets showing the most significant enrichment scores involved down-regulation of genes activated by estradiol in MCF7 breast cancer cells [38], genes involved in cell proliferation (activated by the E2F and Myc transcription factors), and genes encoding ribosomal and protein synthesis factors (Figure 10B and Appendix A). In particular, gene set enrichment plots showed a strong anti-correlation in the gene expression effects of estradiol and tamoxifen (Figure 10C) in keeping with a largely antagonistic action of tamoxifen on estrogen receptor function in luminal breast epithelial cells. Thus, in agreement with previous work, our transcriptome analysis suggests that tamoxifen induces a proliferative arrest of T47D cells by interfering with estrogen receptor transcriptional activity leading to a down-regulation of genes required for proliferation and protein synthesis.

We next compared the gene expression of T47D parental cells with 2 independent H2A.J-KO cell lines in proliferating conditions. We sought genes that were differentially regulated (FDR < 0.05) in a similar manner in the 2 H2A.J-KO cell lines compared to T47D in order to exclude differences in the Crispr/Cas9 H2A.J-KO clones that were unrelated to the H2A.J gene inactivation. 90 genes were down-regulated and 17 genes up-regulated (Figure 11A,B and Appendix A). Interestingly, 39/90 (43%) of the genes down-regulated in the H2A.J-KO cells corresponded to genes that were down-regulated in T47D cells by tamoxifen treatment, and 9/17 (53%) of genes that were up-regulated in the H2A.J-KO cells were up-regulated in T47D by tamoxifen treatment (Figure 11C). Furthermore, 27 of the genes that are differentially expressed in the H2A.J-KO cells are also putative direct targets of *ESR1* (*ADAMTS8*, *APBB2*, *CXCL12*, *FAM189A2*, *FHL2*, *FOS*, *GAS6*, *GRIK3*, *HES1*, *HSPB8*, *LRP4*, *MAPT*, *NDRG1*, *NEDD9*, *NEK10*, *OLFML3*, *PDZK1*, *RBBP8*, *ROBO2*, *SCGB2A1*, *SEL1L3*, *SLC7A5*, *STC1*, *SULF2*, *SULT1A2*, *TFF1*, *TNS1*) according to ChIP-seq data obtained by the Encode consortium and the ChEA transcription factor database [39,40]. These results suggest that H2A.J function is linked to estrogen-regulated gene expression.

We confirmed this RNA-seq differential expression analysis by reverse transcriptase coupled to quantitative PCR experiments to determine the RNA levels for five estrogen-responsive genes (*PDKZ1*, *SPINK5*, *SPINK13*, *SULF2*, and *TFF1*) in the T47D cells versus the two H2A.J-KO cells lines (Appendix A). The expression of these genes was generally lower in the H2A.J-KO lines relative to the parental T47D cells. Expression of a Flag-HA-H2A.J construct in these H2A.J-KO cells led to increased expression of these estrogen-responsive genes (Appendix A). Finally, Western blotting analyses showed that estrogen-induced expression of the TFF1 protein was inhibited in the H2A.J-KO cells relative to the parental T47D cells (Appendix A).

Gene set enrichment analysis of the RNA-seq data strengthened the links between H2A.J and estrogen expression in breast cancer. The most significant gene set was that of genes induced by estradiol in MCF-7 breast cancer cells (the Massarweh_Response_To_Estradiol gene set [41]) (Figure 11D and Appendix A). Genes highly expressed in T47D versus the H2A.J-KO cell lines were highly represented in estradiol-induced genes in MCF-7 cells. Remarkably, genes up-regulated in luminal-B breast cancer (SMID_Breast_Cancer_Luminal_B_Up [42]) relative to other breast cancer types were also enriched for genes highly expressed in T47D versus the H2A.J-KO cell lines. This finding is notable because the *H2AFJ* gene was shown to be amplified and over-expressed in luminal-B breast cancer [7]. These results thus suggest that H2A.J functions in breast epithelial cells to promote the effects of estrogen on gene expression and to increase the expression of genes that are characteristic of luminal-B breast cancer. Finally, the 2 most significant gene sets in the KEGG-Gene Ontology and Hallmark gene sets corresponded to the Cilium_Movement and Epithelial_Mesenchymal_Transition gene sets. Thus, H2A.J in breast epithelial cells also promotes the expression of genes implicated in cancer metastasis.

## 4. Discussion

Histone variants are expressed independently of DNA replication and thus may potentially have increased importance in post-mitotic cells relative to canonical histones. However, relatively little is known regarding possible tissue-specific expression of histone variants. We observed a striking tissue and cell-type specificity for H2A.J expression by immunohistochemical staining of several mouse and human tissues. Some tissues, such as the heart and the brain, showed low levels of H2A.J expression. Our preliminary results for brain tissues suggest that only a small subset of cells, whose identity remains to be established, expresses high levels of H2A.J. In contrast, H2A.J expression was strikingly enriched in the luminal epithelial cells of numerous glands, including mammary, prostate, pancreas, thyroid, stomach, bladder, and salivary glands. The GTEx RNA expression databank for human tissues generally correlated with the protein expression in that testis, prostate, adipose, and breast tissue showed high *H2AFJ* RNA expression whereas brain and heart were low. However, our previous work showed significant accumulation of H2A.J protein in mouse liver and kidney [5], whereas the GTEx data indicate that *H2AFJ* RNA levels are low in human liver and kidney. *H2AFJ* might be expressed at higher levels in mouse liver and kidney compared to humans, or there might be post-transcriptional regulation that accounts for significant H2A.J protein accumulation in these organs despite relatively low RNA levels. We previously noted that H2A.J protein accumulates to higher levels in senescent compared to quiescent human fibroblasts, even though both types of fibroblasts express similar levels of *H2AFJ* RNA [5].

Single-cell RNA-seq data are allowing unprecedented study of cell-type specificities in mammalian tissues. We used published data sets to show that *H2AFJ* RNA expression is enriched in luminal cells of the mammary and prostate glands. Thus, enrichment of H2A.J in luminal epithelial cells is at least partly due to differential expression of its RNA in luminal epithelial cells. Further study of transcriptomics data from breast and prostate cancer showed that *H2AFJ* RNA expression was highest in luminal-type cancers derived from these tissues. *H2AFJ* RNA and H2A.J protein are thus novel biomarkers of luminal-type breast and prostate cancers. When considering all types of breast cancer, high expression of *H2AFJ* RNA is correlated with better survival compared to low levels of *H2AFJ* expression. Since luminal breast cancers tend to have a better prognosis compared to basal breast cancers [35], this survival advantage of high *H2AFJ* expression when considering all breast cancers may simply reflect its association with luminal epithelial cells. We also considered survival versus *H2AFJ* expression levels after separating breast cancer cell types into their PAM50 molecular subtypes. In this instance, high *H2AFJ* expression had no significant effect on the survival of basal-like or luminal-B breast cancers, but was correlated with slightly better survival of HER2+ and normal-like breast cancer, and correlated with worse survival of luminal-A type breast cancers. Overall, luminal-B type breast cancers have intrinsically higher expression of *H2AFJ* and a poorer prognosis compared to luminal-A type breast cancers [7]. We speculate that the poorer survival of luminal-A cancers with the highest expression of *H2AFJ* might reflect a state that is more similar to the luminal-B type cancers. It is more difficult to interpret the slightly better survival of HE2+ and normal-like cancers that express higher *H2AFJ* RNA.

The function of H2A.J in luminal epithelial cells is not clear. We obtained homozygous H2A.J-KO mice that are viable and fertile. Their phenotypic characterization is ongoing, but the ability of mutant females to successfully raise pups indicates that they do not have a strong defect in mammary gland milk production. Likewise, the male fertility suggests that H2A.J is not required for prostate gland function. Our transcriptome analysis of T47D luminal breast cancer cells indicated that knocking out the *H2AFJ* gene reduced the expression of several estrogen-responsive genes and genes whose expression is up-regulated in luminal-B breast cancer, as well as genes involved in the EMT and cilium movement. Thus, H2A.J appears to facilitate the expression of some genes that are likely to contribute to luminal-type breast tumorigenesis. This result is consistent with work showing that *H2AFJ* is in a genomic region that tends to show copy number amplification in luminal B cancers, and the *H2AFJ* gene itself tends to be hypo-methylated and over-expressed in these cancers, leading to the suggestion that it might be a luminal-B oncogene [7]. Further work will be necessary to experimentally test the importance of H2A.J in luminal breast cancer. The availability of the H2A.J-KO mouse will facilitate an evaluation of its role in mouse models for luminal breast cancer [43].

The mechanism by which H2A.J influences gene transcription remains to be determined. The effect of H2A.J on transcription appears to depend on the cell type. Our previous work indicated that H2A.J stimulated inflammatory gene expression in senescent human fibroblasts and others reported a similar role for H2A.J in glioblastoma cells [11]. Interestingly, over-expression of *H2AFJ* is associated with lower survival in Low-Grade Gliomas (Appendix A) which perhaps may be related to its role in stimulating inflammatory gene expression in these cells [11]. However, most inflammatory genes were not highly expressed in T47D cells and *CXCL12* was the only inflammatory gene whose expression was reduced by knock-out of the *H2AFJ* gene in T47D cells. Luminal breast cells express the ESR1 estrogen receptor transcription factor, and a subset of estrogen-responsive genes, many of which are believed to be direct targets of ESR1, showed decreased transcription in the T47D H2A.J-KO cells. It is striking that H2A.J is highly expressed in luminal epithelial cells of a diverse set of glands. These luminal gland cells presumably all have a high secretory potential, as is the case for senescent fibroblasts, but their transcriptional programs are all distinct. Further work should elucidate the regulatory elements that lead to enhanced *H2AFJ* RNA expression in luminal epithelial cells, and how H2A.J in turn influences the transcriptional program of these diverse glands.

## Figures and Tables

**Figure 1 genes-12-01665-f001:**
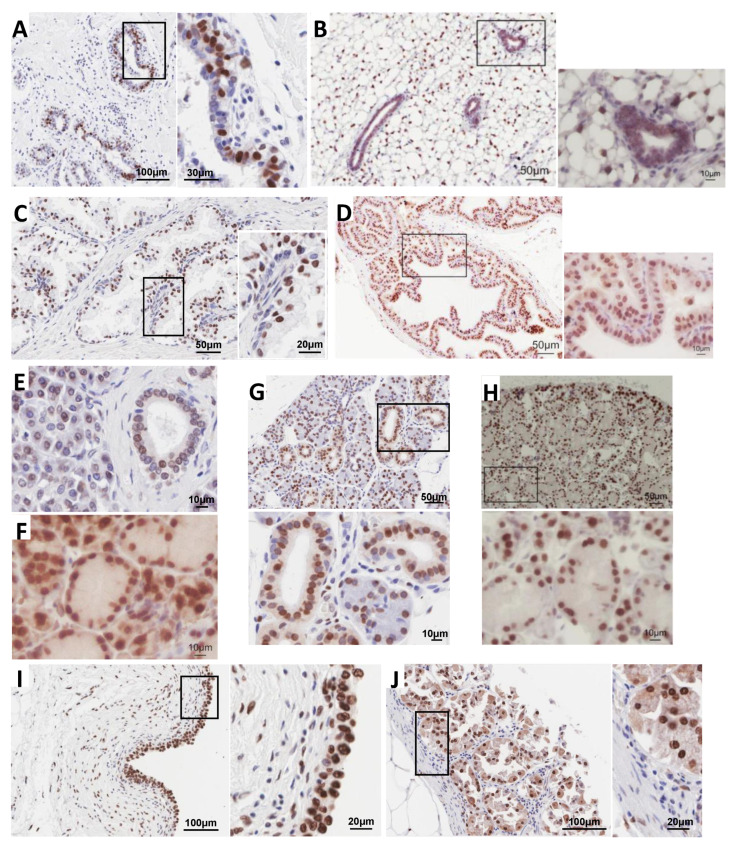
H2A.J is enriched in luminal epithelial cells of multiple glands in humans and mice as seen by IHC analysis of human and mouse tissue sections. Mouse donors were 8–12 weeks of age. (**A**) Human mammary (21-year-old woman). (**B**) Mouse mammary. (**C**) Human prostate (43-year-old man). (**D**) Mouse prostate. (**E**) Human pancreas (50-year-old male). (**F**) Mouse pancreas. (**G**) Human salivary (62-year-old man). (**H**) Mouse salivary. (**I)** Human bladder (35-year-old man). (**J**) Human stomach (45-year-old man). Magnified inserts are derived from the boxed areas.

**Figure 2 genes-12-01665-f002:**
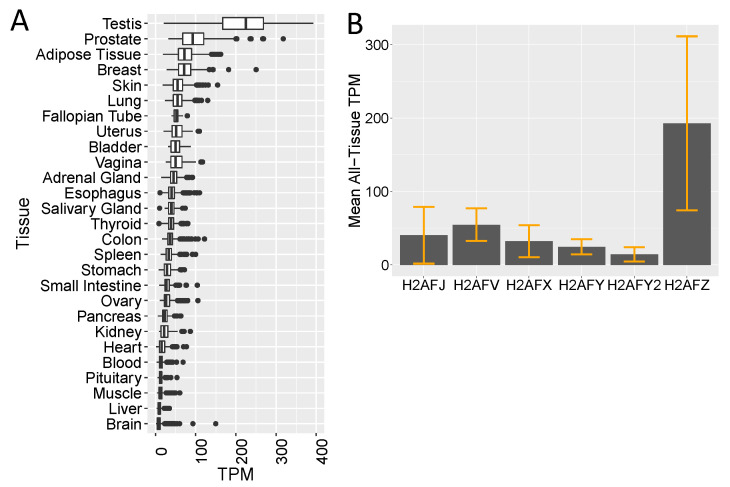
Tissue-specific expression of *H2AFJ* RNA. (**A**) TPM (transcripts per million) data for *H2AFJ* RNA in 27 different human tissues. (**B**) Mean all-tissue TPM for the indicated H2A histone variant genes.

**Figure 3 genes-12-01665-f003:**
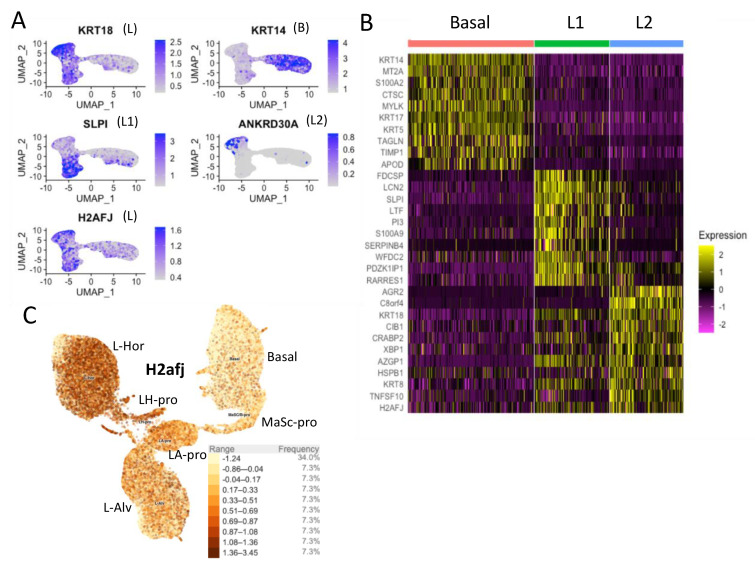
Single-cell RNA expression of human and mouse mammary epithelial cells. (**A**) UMAP representation of RNA expression for the indicated genes in single human mammary gland cells. This UMAP analysis identified 2 luminal-cell clusters (L1, L2) and a basal-cell cluster (**B**). The *H2AFJ* and *KRT18* RNAs are found in both luminal-cell clusters. (**B**) Heat map showing the top 10 genes specific for each human mammary cell cluster along with the *H2AFJ* gene in the last row. (**C**) *H2afj* gene expression in a UMAP clustering of single cells from the mouse mammary gland. This UMAP analysis revealed 6 cell clusters: Luminal-Hormone-sensitive (L-Hor), Luminal-Hormone-sensitive progenitor (LH-pro), Luminal-Alveolar (L-Alv), Luminal-Alveolar progenitor (LA-pro), Basal, and Mammary Stem/progenitor Cell (MaSc-pro). H2afj RNA is enriched in the luminal cell clusters with the highest levels in the Luminal-Hormone-sensitive cluster.

**Figure 4 genes-12-01665-f004:**
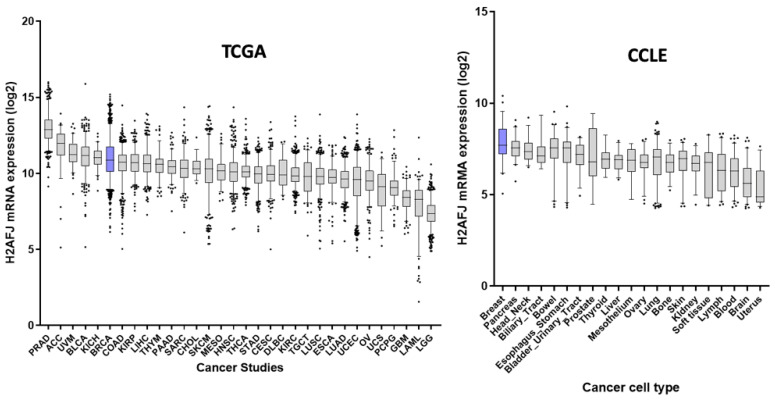
*H2AFJ* is highly expressed in luminal cancer cells. **Left panel**: Log2 of *H2AFJ* RNA levels from transcriptomic data in the Cancer Genome Atlas (TCGA) for prostate adenocarcinoma (PRAD), adrenocorticol carcinoma (ACC), breast carcinoma (BRCA), uveal melanoma (UVM), bladder carcinoma (BLCA), kidney chromophobe carcinoma (KICH), liver hepatocellular carcinoma (LIHC), kidney renal papillary cell carcinoma (KIRP), colon/rectal adenocarcinoma (COAD), thymoma (THYM), skin cutaneous melanoma (SKCM), sarcoma (SARC), pancreatic adenocarcinoma (PAAD), cholangiocarcinoma (CHOL), head and neck squamous cell carcinoma (HNSC), mesothelioma (MESO), diffuse large B-cell lymphoma (DLBC), thyroid carcinoma (THCA), cervical squamous cell carcinoma and endocervical adenocarcinoma (CESC), testicular germ cell tumors (TGCT), stomach adenocarcinoma (STAD), kidney renal clear cell carcinoma (KIRC), lung squamous cell carcinoma (LUSC), uterine corpus endometrial carcinoma (UCEC), esophageal carcinoma (ESCA), ovarian serous cystadenocarcinoma (OV), lung adenocarcinoma (LUAD), pheochromocytoma and paraganglioma (PCPG), uterine carcinosarcoma (UCS), glioblastoma (GBM), acute myeloid leukemia (LAML), brain lower grade glioma (LGG). **Right panel**: Log2 of *H2AFJ* RNA levels from transcriptomic data of cancer cell lines from the Broad Institute Cancer Cell Line Encyclopedia (CCLE). Tissues and cell lines of breast origin are shown in blue.

**Figure 5 genes-12-01665-f005:**
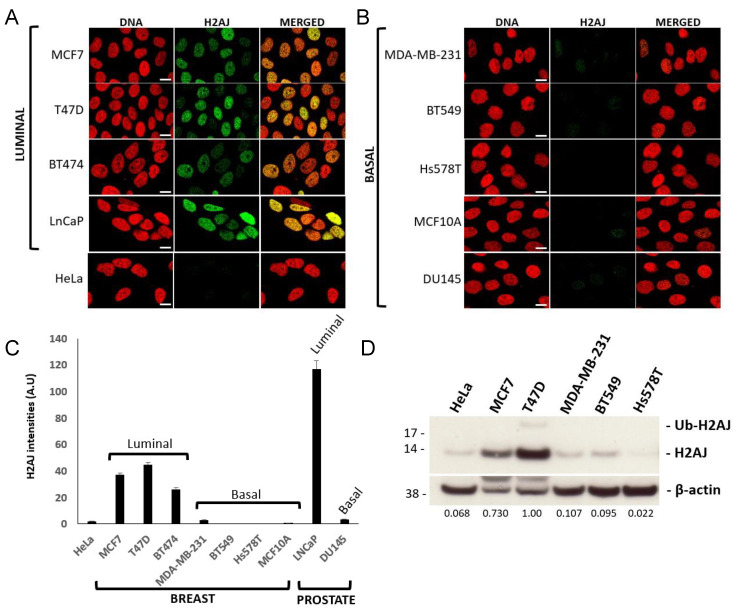
H2A.J protein is expressed at higher levels in breast and prostate cancer cell lines of luminal versus basal origin. (**A**,**B**) Immunofluorescent images with H2A.J-specific antibodies. All images were taken with the same exposure conditions. Scale bar = 10 µm. (**A**) Luminal breast (MCF7, T47D, BT474) and prostate (LnCAP) cancer cell lines [30,31]. HeLa cervical cancer cells were found to express low-levels of H2A.J relative to luminal cancer cells. (**B**) Basal or non-luminal breast (MDA-MB-2341, BT549, Hs578T, MCF10A) and prostate cancer cell lines [30,31,32]. (**C**) Quantification of the H2A.J signal intensities in the images shown in (**A**,**B**). Error bars represent standard errors of the mean. (**D**) H2A.J Western blot of the indicated cancer cell lines. ß-actin was used as a loading control. The numbers under the panels represent the intensity ratios for H2A.J/ß-actin signals normalized to the T47D ratio. Ub-H2AJ refers to a putative ubiquitinated form of H2A.J.

**Figure 6 genes-12-01665-f006:**
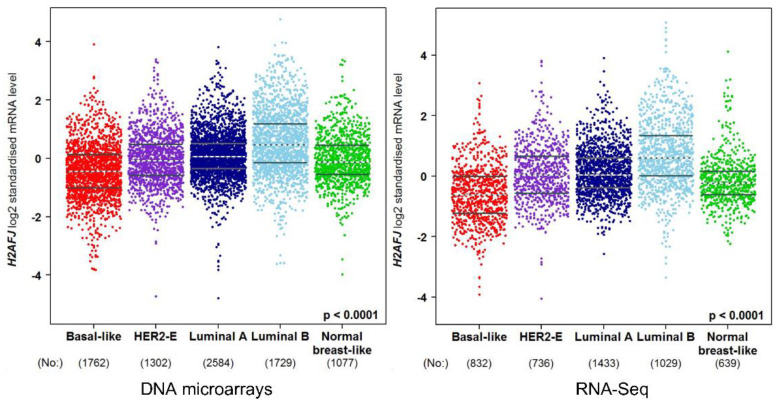
Relative levels of H2AFJ RNA (log2 standardized mRNA levels) in the different PAM50 classes of breast cancer. The left panel shows transcriptomic data obtained by DNA microarrays whereas the right panel shows transcriptomic data obtained by RNA-seq. The number of cancers analyzed in each class is indicated at the bottom of each cancer class. Each point represents a cancer sample. The plots were generated by bc-GenExMiner [33]. The *p*-value is the Welch analysis of variance statistic for a significant difference between the means of all groups. The Dunnett–Tukey–Kramer’s test for pairwise multiple comparisons of the means for each group indicated that Luminal-B > Luminal-A, Luminal-A > HER2-E, Normal-like, and Basal-like, and HER2-E and Normal-like > Basal-like with *p* < 0.0001.

**Figure 7 genes-12-01665-f007:**
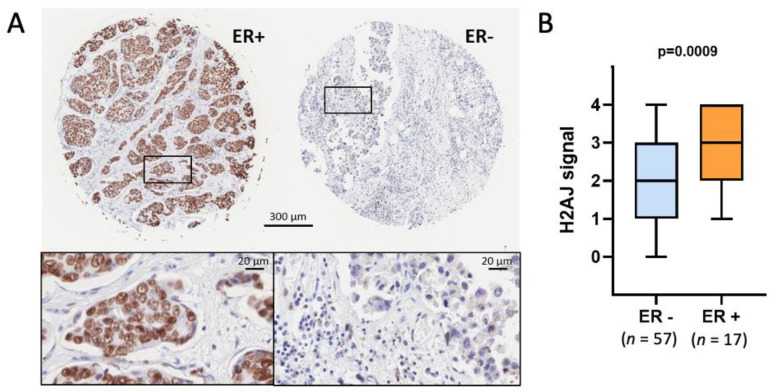
H2A.J is expressed at higher levels in luminal ER-positive breast cancer. (**A**) Representative examples of H2A.J IHC of ER+ and ER- breast cancer biopsies from a breast cancer tissue microarray. The complete microarray is shown in Appendix A. (**B**) Quantification of the H2A.J signals from 57 ER-negative breast cancer samples and 17 ER-positive samples. Statistical significance was determined by a two-tailed, unpaired Student’s *t*-test.

**Figure 8 genes-12-01665-f008:**
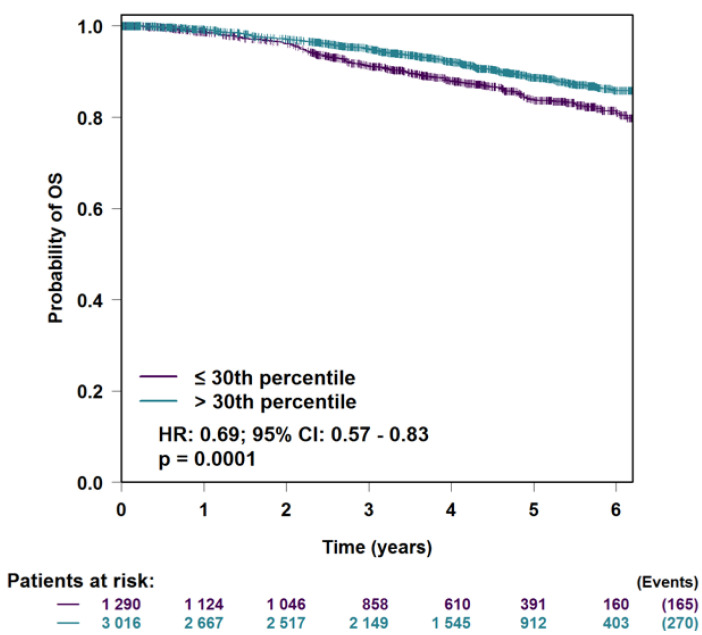
Kaplan–Meier survival curves for the probability of overall survival (OS) for all breast cancers showing the highest 30th percentile of *H2AFJ* expression versus those showing the lowest 30th percentile of *H2AFJ* expression based on RNA-seq data. The total number of patients at risk in each class for each time point is shown below the graph along with the total number of events (deaths) in each cohort at the final time point. The hazard ratio (HR) and its confidence intervals (CI) are shown with the significance *p*-value. The curves were calculated with bc-GenExMiner v4.7 [27] using the optimal splitting criterion on breast cancer cohorts from TCGA and the Swedish Cancerome Analysis Network-Breast (SCAN-B) Initiative (GSE96058 and GSE81538).

**Figure 9 genes-12-01665-f009:**
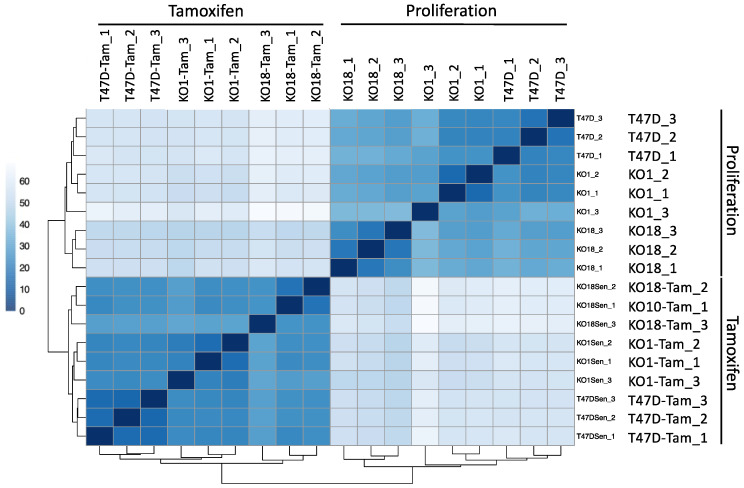
Euclidean distance heat map of the RNA-seq data for the indicated cell lines and experimental conditions.

**Figure 10 genes-12-01665-f010:**
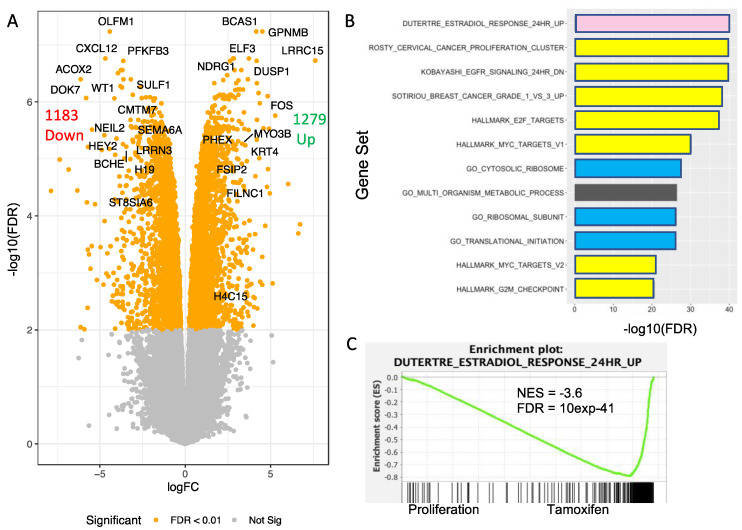
Genes differentially expressed upon treatment of T47D cells with tamoxifen. (**A**) Volcano plot of log fold-change (logFC) versus -log10 of the false discovery rate (FDR) for genes differentially expressed in proliferating versus tamoxifen-treated T47D cells. Orange dots represent genes with FDR < 0.01. The indicated genes are those with log2 fold changes of +/− 3 and an average expression of greater than 1 in the limma–voom analysis. (**B**) Enrichment analysis indicating significant correlations of genes differentially expressed upon tamoxifen treatment with gene sets involved in estradiol response (pink), breast cancer or cell proliferation (yellow), protein synthesis (blue), and metabolic process (black). (**C**) Enrichment plot of tamoxifen-regulated genes compared to the c2 gene set “Dutertre_Estradiol_Response_24HR_UP” [37,38]. NES (normalized enrichment score).

**Figure 11 genes-12-01665-f011:**
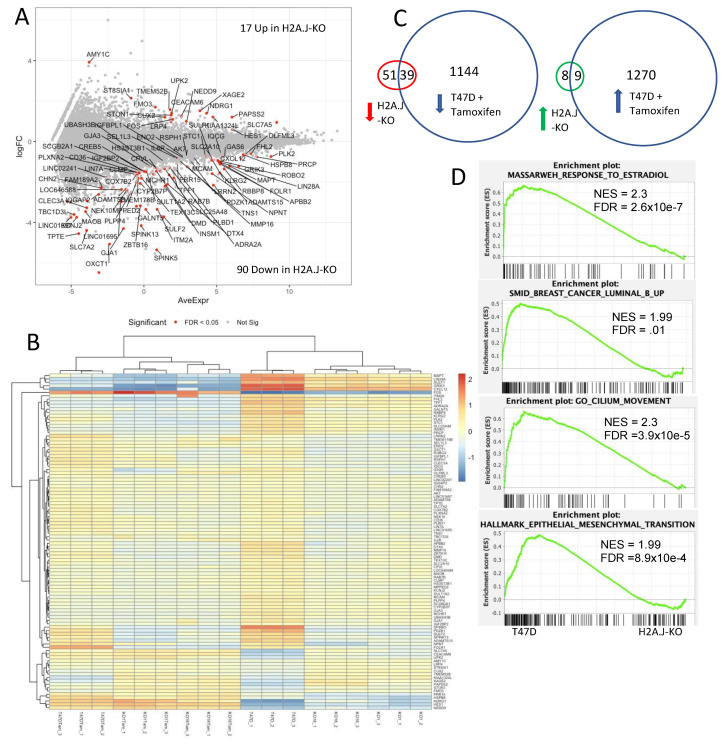
Genes differentially expressed in H2A.-KO cell lines versus parental T47D cells. (**A**) Log2 fold-change (logFC) versus average gene expression for the differentially-expressed genes. The 90 genes down-regulated in H2A.J-KO cells relative to T47D and the 17 genes up-regulated in the H2A.J-KO cells with false-discovery rate (FDR) < 0.05 are indicated in red with their gene names. (**B**) Heat map of the differentially expressed genes for all samples. (**C**) Venn diagrams showing overlaps in gene expression affected by the H2A.J-KO and tamoxifen-treated T47D cells. (**D**) Gene sets in the c2 and Hallmark MsigDB [37] that are enriched with regards to the transcriptome of H2A.J-KO cells compared to parental T47D cells. NES (normalized enrichment score).

## Data Availability

The RNA-seq fastq files have been deposited at the EBI Array Express Database under accession number E-MTAB-10861. Bioinformatic analyses have been deposited at Zenodo (doi:10.5281/zenodo.5414141, doi:10.5281/zenodo.5519652, doi:10.5281/zenodo.5517186).

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
