# Peer review of "Histone Variant H2A.J Is Enriched in Luminal Epithelial Gland Cells"

_genes, 2021, doi:10.3390/genes12111665_

Round 1

Reviewer 1 Report

I suggest improving the introduction , and I suggest putting in the supplementary figures the legends.

Author Response

Response to Reviewer 1 Comments

Point 1: I suggest improving the introduction , and I suggest putting in the supplementary figures the legends.

Response 1: We have improved the Introduction to make it more accessible to non-specialists. We have also added the legends to the Supplementary Figures as requested.

Reviewer 2 Report

Redon et al have studied the expression of Histone variant H2A.J and found it more abundant in luminal subtype of breast and prostate cancer. The study has its novelty and is interesting. I just have minor considerations/questions:

  • There are p-values in Fig. 6 that is not clear to what comparison they belong to.
  • Fig. 7: why only showing 2 samples out of 58 + 18 samples? 
  • Fig. 1, 5, 7: how specific is the antibody? what are the positive and negative controls and could you show those ctrl images as well?
  • Would you consider presenting a table including the patient clinical data (or citing it if already published) and performing statistical analysis between your IHC scoring and clinical data?
  • line 337 and 355: "Breast Cancer Survival" is not a function but improving it could be an effect/role ....

Thanks                                                                     

Author Response

Point 1: There are p-values in Fig. 6 that is not clear to what comparison they belong to.

Response 1: The p-value is the Welch analysis of variance statistic for a significant difference between the means of all groups. The Dunnet-Tukey-Kramer’s test for pairwise multiple comparisons of the means for each group indicated that Luminal-B > Luminal-A, Luminal-A > HER2-E, Normal-like, and Basal-like, and HER2-E and Normal-like > Basal-like with p < 0.0001. We have added this information to the Figure 6 legend.

Point 2: Fig. 7: why only showing 2 samples out of 58 + 18 samples? 

Response 2: We showed 2 representative samples in Fig. 7 at high magnification to allow readers to judge the quality of the staining, and the quantification of the staining  gave the statistical analysis for all samples. However, in response to this reviewer's comment, we have added a new Supplementary Figure 5 with an image representing the staining of the entire microarray containing all breast cancer core biopsies. 

Point 3Fig. 1, 5, 7: how specific is the antibody? what are the positive and negative controls and could you show those ctrl images as well?

Response 3: For the mouse tissue immunohistochemistry, we showed positive (WT) and negative (congenic H2A.J-KO) controls in Supplementary Figure 1. We have also added to Supplementary Fig. 1 an example of competition of the H2A.J signal by an excess of the C-terminal H2A.J-specific peptide that was used as an immunogen to prepare the antibody. For the human cells, we show in Supplementary Fig. 9 the presence of H2A.J staining in T47D cells and its absence in 2 different H2A.J-KO derivatives. We have also previously published an example of peptide competition of H2A.J staining of human skin sections in Rübe et al. 2021 that we cite. As we note in the text, we have previously published extensive characterisation of the specificity of the this antibody in Contrepois et al. Nature Communications 2017 and in Rübe et al. 2021.

Point 4Would you consider presenting a table including the patient clinical data (or citing it if already published) and performing statistical analysis between your IHC scoring and clinical data?

Response 4: We have added an additional Supplementary Fig. 5 where we show the H2A.J staining for the complete breast cancer microarray that we bought from US Biomax. In the supplementary figure legend, we have added the available clinical data for these samples that were provided with this tissue microarray by US Biomax. In Figure 7B, we provide the statistical analysis for IHC scoring relative to ER status. The remaining clinical classes were present in too small numbers to allow us to draw any conclusions.

Point 5line 337 and 355: "Breast Cancer Survival" is not a function but improving it could be an effect/role ....

Response 5: We intended this to mean Breast cancer survival as a mathematical function of survival as calculated by Kaplan-Meier curves. However, to remove all ambiguity, we have changed the section subtitle to "H2AFJ Expression in Relation to Breast Cancer Survival".